# The Impact of Physical Activity on Mental Health during COVID-19 Pandemic in China: A Systematic Review

**DOI:** 10.3390/ijerph19116584

**Published:** 2022-05-28

**Authors:** Mengfei Li, Qianhui Wang, Jing Shen

**Affiliations:** Department of Physical Education, China University of Geosciences (Beijing), No. 29, Xueyuan Road, Haidian District, Beijing 100083, China; lmflyh@163.com (M.L.); qianhui0809@163.com (Q.W.)

**Keywords:** COVID-19, physical activity, mental health, anxiety, depression, China

## Abstract

The outbreak and spread of novel coronavirus disease (COVID-19) in 2019 was a public health emergency of global concern. As an important health behavior, physical activity (PA) and its impact on mental health have been increasingly explored during the epidemic period. The keywords and references were searched on PubMed, Web of Science, Scopus, Cochrane Library, EBSCO, and CNKI since the inception of an electronic bibliographic database until October 2021. A total of 2979 articles were identified, of which 23 were eligible for inclusion to examine the relationship between PA and mental health during the COVID-19 epidemic. Residents with regular PA, high-intensity PA, and PA duration of 30–60 min or more per day were associated with a lower risk of anxiety, depression, and negative emotions. In contrast, residents with no exercise and physical inactivity were more likely to have anxiety, depression, sleep disturbances, and lower subjective well-being. In addition, the dose–response curve between PA and negative emotions indicated a U-shaped relationship, and there were urban–rural differences in the relationship between emotional status and the levels of PA in adolescents. The findings have significant implications for addressing mental health issues during the current pandemic and future pandemics. Future studies adopting an experimental study design, conducting objective PA measures, and focusing on the vulnerable subpopulations are warranted to further explore the association of PA on mental health during the COVID-19 pandemic.

## 1. Introduction

The novel coronavirus (COVID-19) pandemic began in December 2019 and, as a result of its extremely high rate of infection, spread rapidly in a short period of time, resulting in a pandemic that has severely affected most countries. With the international spread of epidemics posing a major public health risk to the world, the global outbreak of COVID-19 of international concern was officially declared a public health emergency by the World Health Organization (WHO) on 30 January 2020 and upgraded to a pandemic on 11 March. In response to the sudden outbreak, the Chinese government has adopted a series of effective measures, including community closure, home isolation, and social distancing, to halt the spread of the pandemic and to minimize the possibility of transmission and infection of COVID-19.

These measures, as well as the outbreak itself, brought inevitable mental health consequences, such as stress, fear, or anxiety. Xiong reported elevated rates of mental health outcomes (i.e., 50.9% for anxiety, 48.3% for depression, 53.8% for post-traumatic stress disorder, 38% for psychological distress, and 81.9% for stress) in the global general population during the outbreak [1]. In the systematic review by Samji et al., negative mood or worsening of mental health increased as a consequence of COVID-19 pandemic control measures. Among them, older children and adolescents in Germany showed more depressive symptoms due to reduced social exposure during the pandemic [2]. Since before the pandemic, the number of reported symptoms of depression and anxiety has increased and the overall mental health status has deteriorated [3].

In fact, prior to the pandemic, physical activity (PA) had been identified as a protective factor against anxiety [4]. Moreover, some studies have shown that maintaining a certain amount of PA during the epidemic can decrease anxiety, depression, and other symptoms [5]. Increasing PA may reduce the negative psychological effects of confinement and thus improve mental health [6]. However, PA levels have been observed to decrease in the general population in several countries [7,8].

It is reported that there is a certain correlation between the decline in PA and the occurrence of mental health problems [9,10]. The increase in PA was associated with improving mental health [9,11]. However, previous reviews focused on other countries, but not China. The reason for this study is that China has a vast territory and a large population base, but the distribution and density of the population are uneven, and the living environments and habits between rural and urban areas are very different. Furthermore, China’s epidemic prevention and control policies have been timely and stringent to contain the further spread of the virus, which can serve as a reference for other countries in the world. Therefore, we conducted this systematic review to examine the association between PA and mental health during the COVID-19 pandemic in China.

## 2. Materials and Methods

### 2.1. Study Selection Criteria

Studies that met all of the following criteria were included in the review: (1) study design: cross-sectional studies, or longitudinal studies; (2) study subjects: healthy people of all ages; (3) study time frame: during the COVID-19 pandemic; (4) exposure: PA-related behaviors; (5) outcome: mental health (e.g., anxiety, depression, well-being, emotional state); (6) article type: peer-reviewed publications; (7) time window of search: from the inception of an electronic bibliographic database to October 12, 2021; (8) language: articles written in English or Chinese; (9) country: China.

Studies that met any of the following criteria were excluded from the review: (1) studies which had no outcomes related to mental health status; (2) studies which had no PA measures; (3) studies which did not examine the relationship between PA and mental health during the COVID-19 epidemic; (4) studies which were not conducted in China; (5) studies that included participants who were pregnant women, people with disabilities, or patients; and (6) editorials, study/review protocols or review articles.

### 2.2. Search Strategy

A keyword search was performed in five electronic bibliographic databases: PubMed, Web of Science, Scopus, Cochrane Library, EBSCO, and CNKI (a central Chinese scientific literature database). The search algorithm included all possible combinations of keywords from the following three groups: (1) “coronavirus”, “SARS-CoV-2”, “COVID-19”, “COVID-19”, “COVID19”, “COVID 19”, “COVID-2019”, “2019nCoV”, “2019-nCoV”, “2019 nCoV”, “SARSCoV-2”, “SARS-CoV-2”, “SARS-COV2”, “SARS CoV2”, “coronavirus”, “nCoV-19”, “corona virus”, “lockdown”, “social isolation”, “quarantine”, “stay-at-home”, “stay at home”, “staying home”, “stay home”, “staying at home”, “home confinement”; (2) “motor activity”, “motor activities”, “sport”, “sports”, “physical fitness”, “physical exertion”, “physical activity”, “physical activities”, “physical inactivity”, “sedentary behavior”, “sedentary behaviour”, “sedentary behaviors”, “sedentary behaviours”, “sedentary lifestyle”, “sedentary lifestyles”, “inactive lifestyle”, “inactive lifestyles”, “exercise”, “exercises”, “active living”, “active lifestyle”, “active lifestyles”, “outdoor activity”, “outdoor activities”, “walk”, “walking”, “running”, “bike”, “biking”, “bicycle”, “bicycling”, “cycling”, “stroll”, “strolling”,, “active transport”, “active transportation”, “active transit”, “active commuting”, “travel mode”, “physically active”, “physically inactive”; and (3) “mental health”, “mental disorders”, “psychology”, “emotions”, “mental health”, “mental illness”, “mental disorder”, “mental disorders”, “well-being”, “wellbeing”, “mood”, “anxiety”, “depression”, “depressive”, “distress”, “stress”, “resilience”, “sleep”, “insomnia”, “affective”, “fear”, “phobia”, “emotion”, “emotions”, “emotional”, “psychological”, “psychology”, “psychiatry”, “psychiatric”. The MeSH terms “coronavirus”, “SARS-CoV-2”, “COVID-19”, “exercise”, “mental health”, “mental disorders”, “psychology”, “emotions”, “humans”, or “China” were used in the PubMed search. Title and abstract screening were conducted on the articles identified from the keyword search. Potentially eligible articles were retrieved, and their full texts were evaluated. Two coauthors of this review independently performed title and abstract screening. Cohen’s kappa (κ = 0.80) was used to assess inter-rater agreement. Discrepancies between the two authors were resolved through discussion.

### 2.3. Data Extraction and Synthesis

A standardized data extraction form was used to collect the following methodologic and outcome variables from each article, including authors, publication year, city or region or country, study design, sample size, age range, proportion of female participants, sample characteristics, statistical model, non-response rate, setting, type of PA measure, detailed measure of PA, types of mental health measure, effects of PA on mental health, and main findings of the relationship between PA and mental health.

### 2.4. Study Quality Assessment

We used the National Institutes of Health’s Quality Assessment Tool for Observational Cohort and Cross-Sectional Studies to assess the quality of each included study [12]. This assessment tool rates each study based on 14 criteria. For each criterion, a score of 1 was assigned if “yes” was the response, whereas a score of 0 was assigned otherwise (i.e., if an answer of “no”, “not applicable”, “not reported”, or “cannot determine” resulted). A study-specific global score ranging from 0 to 14 was calculated by summing the scores across all criteria. The study quality assessment helped measure the strength of scientific evidence but was not used to determine the inclusion of studies. Two coauthors of this review independently conducted the study quality assessment, with discrepancies resolved through discussion with a third coauthor.

## 3. Results

### 3.1. Identification of Studies

Figure 1 shows the study selection flowchart. We identified a total of 2979 articles through keyword and reference search, including 612 articles from PubMed, 1645 articles from Web of Science, 361 articles from Cochrane Library, 130 articles from Scopus, 39 articles from EBSCO, and 192 articles from CNKI. After removing duplicates, 1923 unique articles underwent title and abstract screening, after which 1881 articles were excluded. The full texts of the remaining 42 articles were reviewed against the study selection criteria. Of these, 19 articles were excluded. The primary reasons for exclusion were lack of PA measures, not reporting the relationship between PA and mental health, not being conducted in China, or the participants included patients, pregnant, or suspected COVID-19 patients. The remaining 23 studies that examined the association of PA and mental health during the COVID-19 epidemic were included in the present review.

### 3.2. Basic Characteristics of the Included Studies

Table 1 summarizes the basic characteristics of the 23 included studies. A total of 11 studies were conducted across provinces in China: two studies each in Wuhan, Yanan and Hechi, one each in Guiyang, Nanjing, Fuzhou, and Wuhu, one in Nanjing and Suzhou, and the remaining one in the south of China. A total of 18 studies were published in 2020 and the remaining 5 in 2021, but all 5 studies were conducted in 2020. All articles adopted a cross-sectional study design except two [13,14], which adopted a longitudinal study design. Sample sizes were generally large but varied substantially across studies. One study had a sample size less than 99; four had a sample size between 100 and 999; the vast majority of studies (*n* = 17) had a sample size between 1000 and 9999; it is worth mentioning that there was a study with a sample size of more than 10,000. The mean and median sample sizes were 3157 and 1608, respectively, with a standard deviation of 3196 and a range from 66 to 12,107. A total of nine studies recruited children and adolescents, nine studies recruited college students (two of these were studies on returning college students) and the remaining five recruited adult residents. All included studies recruited both males and females. Among the 22 articles that reported gender distribution, 12 studies indicated that women accounted for over half (51–100%) of the analytic sample and 10 studies indicated that women accounted for less than half (35–50%) of the analytic sample.

Table 2 summarizes the PA and mental health measures in the included studies. A total of 22 studies adopted a subjective PA measures questionnaire reported by participants themselves. The remaining study adopted both objective measures of obtained PA data from WeChat’s pedometer and subjective measures. Self-reported PA questionnaires included both standardized questionnaires (e.g., the International Physical Activity Questionnaire (IPAQ), IPAQ—Short Form, Leisure-Time Exercise Questions) and self-compiled questionnaires. Among these subjective PA measures studies, 12 studies used questionnaires known to be validated in previous studies. Mental health outcomes included anxiety (*n* = 12), depression (*n* = 14), and other variables of psychological health (e.g., stress, well-being, emotional states, sleep disorders) (*n* = 13). Anxiety measures adopted GAD-7 (*n* = 5), SAS (*n* = 4), DASS-21 (*n* = 2), and Screen for Child Anxiety Related Disorders (n = 1). Depression measures adopted PHQ-9 (*n* = 4), SDS (*n* = 2), CES-D (*n* = 3), DASS-21 (*n* = 2) and Depression Self-Rating Scale for Children (*n* = 1). Other variables of mental health measures mainly adopted DASS-21 (*n* = 2), POMS (*n* = 4), YSIS (*n* = 2), FCV-19S (*n* = 2), and GWS (*n* = 1).

### 3.3. Key Findings

Table 3 summarizes the effect of PA on mental health among Chinese residents during the COVID-19 epidemic.

#### 3.3.1. Association of PA and Anxiety during the COVID-19 Epidemic

Eleven studies examined the associations of PA and anxiety during the COVID-19 epidemic. Among them, seven studies reported statistically significant associations between PA frequency and anxiety [6,15,16,17,19,29,32]. For example, Chen et al. reported that physical exercise 1–2 times a week, 3–4 times a week, 5–6 times a week, and more than 6 times a week (e.g., the estimated ORs ranged from 0.49 to 0.64) are the influencing factors of psychological anxiety for college students [16]. Wu et al. reported physical exercise 1–2 times a week is negatively associated with college students’ anxiety (OR = 0.67) [32]. Deng et al. reported that exercising more than 1 to 2 times/week was associated with lower anxiety scores [6]. Moreover, regular exercise was associated with lower anxiety scores [6,15,29]. By contrast, residents with no exercise were more likely to have anxiety [17,19].

Two studies reported a statistically significant association between PA duration and anxiety [14,21]. Lu et al. reported that high PA time and low sitting time are associated with a lower prevalence of anxiety symptoms [14]. Chen et al. reported exercise duration of 30–60 min/day and ≥60 min/day were associated with a lower risk of anxiety [21].

Two studies reported statistically significant associations between PA intensity and anxiety [25,30]. Chi et al. reported that being moderately and highly active physically was significantly associated with a lower level of anxiety symptoms [30]. Xiang et al. reported that a high level of PA was significantly associated with low anxiety, while a moderate level of PA showed insignificant association with anxiety symptoms [25].

#### 3.3.2. Association of PA and Depression during the COVID-19 Epidemic

Fourteen studies focused on associations between PA and depression.

Six studies reported statistically significant associations between PA frequency and depression [6,15,17,19,26,32]; Wu et al. reported that physical exercise 1–2 times a week (OR = 0.73) and 3–4 times a week (OR = 0.65) is negatively associated with depression in college students [32]. Zhang et al. reported that exercise more than 1 time a week was significantly associated with epidemic-related depression [26]. Regular physical exercise was associated with depression significantly [6,15]. Exercised >1 h/week was associated with lower scores on depression [6].

By contrast, Lin et al. reported those exercising regularly but lasted less than 2 months were more likely to have mild depression [19]. No exercise was more likely to give rise to depression [17,19].

Three studies reported statistically significant associations between PA duration and depression [14,21,28]. Lu et al. reported high PA time was associated with a lower risk to develop depressive symptoms [14]. Exercise duration of 30–60 min/day (OR = 0.64) and ≥60 min/day (OR = 0.67) were associated with a lower risk of depression [21]. By contrast, physical exercise duration of <30 min/day (OR = 1.641) was more likely to give rise to depression [28].

Five studies reported statistically significant associations between PA intensity and depression [13,20,25,27,30]. Zhang et al. reported PA significantly alleviated depression, with each 100-unit increase in METs of total PA corresponding to a change of −0.04 (95%CI: −0.080, −0.0022) in the depression score [13]. Moderate and high levels of PA were significantly associated with reduced depression [25,27,30]. Lin et al. found moderate-intensity PA was significantly negatively correlated with depression, while vigorous-intensity PA showed an insignificant association [20].

#### 3.3.3. Association of PA and Other Variables of Mental Health during the COVID-19 Epidemic

Thirteen studies reported the association of PA and other variables of mental health (e.g., stress, subjective well-being, emotional state, insomnia symptoms, affect, and mood).

First, PA participation was negatively associated with perceived stress levels [23]. Regular exercise, maintained their exercise habits during the outbreak, exercise frequency of >1 to 2 times/week, exercise duration of >1 h/week, and exercise intensity of >2000 average pedometer steps/day were associated with lower stress scores [6]. In addition, boys’ exercise participation behaviors (such as exercise time, exercise frequency and regular exercise) are better than girls’ [19]. However, Zhang et al. reported that PA was insignificantly associated with stress alleviation [13].

Second, highly active physically [30], and both high PA time and low sitting time (OR = 0.40) [14], were associated with a lower level of insomnia symptoms. By contrast, no physical exercise (OR = 1.85) was significantly associated with sleep disorder [17].

Third, physical exercise was positively correlated with subjective well-being and was significantly higher in women than in males [22]. Inadequate leisure-time PA (OR = 1.16) was associated with a higher risk of lower subjective well-being [18].

Fourth, PA was associated with the mood states. Kang et al. reported that higher PA was significantly associated with lower levels of negative mood states [31]. Xiao et al. reported that PA duration of ≥150 min/week was significantly associated with a lower risk of negative mood, and PA participation was significantly associated with a lower mood disturbance score [33]. Lower PA levels indicated higher negative mood states scores [27]. When focusing on subpopulation, Li et al. found that PA level was significantly negatively associated with total score of emotional state for adolescents in urban areas while there was an insignificant correlation for adolescents in rural areas [9].

In addition, Zhang et al. reported that the dose–response curve between PA and negative emotions indicated a U-shaped relationship [13].

#### 3.3.4. Effect of PA Changes on Mental Health in Different Populations during the COVID-19 Epidemic

In terms of gender subgroup, boys’ exercise participation behaviors (such as exercise time, exercise frequency and regular exercise) are higher than girls’ [19]. Females had spent less time on moderate- and vigorous-intensity PA than males [20]. Studies involving gender contrast all indicated that the mental health of female participants was more susceptible. For example, female participants were reported with higher levels of insomnia, depressive and anxiety symptoms than their male counterparts [30]; women are more likely to have dizziness, headache, fatigue and other physical symptoms due to the epidemic [26]; and female college students are more anxious than boys during the epidemic [29].

In terms of age subgroup, five studies, focusing on children and adolescents, indicated that the epidemic had a negative impact on PA. Teenagers with the pressure about study are more likely to have symptoms of anxiety and depression [14,31]. Children who were left behind showed more severe symptoms of insomnia, depression and anxiety during the epidemic [30].

For the older population, it was reported that there was a high awareness of health management behavior among the Chinese elderly population, with a relatively low proportion of people aged 55–59 and older with less physical activity and less screening time during family isolation [24].

### 3.4. Study Quality Assessment

Appendix A reports criterion-specific and global ratings from the study quality assessment, and is shown in the Appendix A. The included studies on average scored 7 out of 14, with a range from 4 to 11. All included studies of the review clearly stated the research question/objective, specified and defined the study population, had a participation rate of at least 50%, recruited participants from the same or similar populations during the same time period, and prespecified and uniformly applied inclusion and exclusion criteria to all potential participants. The majority of studies implemented valid and reliable outcome measures (*n* = 18) and examined different levels of the exposure in relation to the outcome (*n* = 16); 12 studies implemented valid and reliable exposure measures. Nine studies measured and statistically adjusted key potential confounding variables for their impact on the relationship between exposures and outcomes. In contrast, none of the studies had the outcome assessors blinded to the exposure status of the participants. Only one study had an attrition rate of 20% or less, measured exposures of interest before the outcomes, assessed the exposures more than once during the study period, and had a reasonably long follow-up period that was sufficient for changes in outcomes to be observed. Two studies provided a sample size justification using power analysis.

## 4. Discussion

This study systematically reviewed the literature on the association between PA and mental health during the COVID-19 pandemic in China. A total of 23 studies, including 21 cross-sectional, and 2 longitudinal, were included in this review. The findings from this review demonstrate that residents with regular PA, high-intensity PA, and PA duration of 30–60 min or more per day were associated with a lower risk of anxiety, depression, and negative emotions. In contrast, residents with no exercise and physical inactivity were more likely to have anxiety, depression, sleep disturbances, and lower subjective well-being. In addition, the relationship between PA intensity and emotional efficacy presents an inverted U-shaped curve, that is, both low-intensity and high-intensity physical activity are not conducive to the generation of motor emotional benefit, while the emotional benefit of moderate-intensity activity is the best [34].

Physical isolation during the pandemic has affected normal physical activities, such as the suspension of all sports and fitness facilities, the possibility of participating in group sports activities blocked by home isolation, and the wearing of masks has affected individual outdoor activities. Therefore, barriers to PA exacerbated the negative impact on mental health during the pandemic. It is worth noting that the impacts had gender differences and group differences. Compared to men, women’s PA levels are more susceptible to the influence of the outbreak. The reasons for the difference were that women are more susceptible to external influential factors, and more fearful in the face of the COVID-19 crisis [32]. In addition, as a special group, college students, whose psychological development is still not mature, appeared to have higher anxiety than other groups during the COVID-19 pandemic [35]. This is also mainly due to the fact that college students are more susceptible to information about the transmission route, vulnerable groups and cure rate of COVID-19 [36].

There is growing scientific evidence that PA is a self-selected lifestyle behavior, an effective means of promoting mental health and resisting the risk of the COVID-19 pandemic [37]. Although PA participation has been extensively affected by the COVID-19 epidemic breakout, the importance of its health benefits should not be ignored.

In terms of policy implications, it is necessary to formulate public health policies related to PA promotion during the pandemic to improve mental health, such as guiding and encouraging workouts at home, exercises in outdoor natural environments which help to maintain the social distance, and online PA instruction and interaction. Moreover, special attention should be given to vulnerable groups in terms of mental health (e.g., women, students, and the elderly). By planning and designing PA measures that can be actively implemented in any context, unnecessary negative effects on mental health can be avoided during any pandemic in future.

To our knowledge, this study is the first that systematically reviews the existing literature regarding the association of PA and mental health during the COVID-19 pandemic in China. The review not only considered several domains of mental health (e.g., anxiety, depression, and mood), but also considered PA frequency, duration, and intensity. Understanding the potential impact of PA on mental health during the COVID-19 pandemic has important implications for the current pandemic and future pandemics.

Several limitations pertaining to this review and the included studies should be noted. First, the majority of studies adopted a cross-sectional study design, which excludes a causal interpretation regarding the impact of PA on mental health during the COVID-19 pandemic; therefore, it was only clarified the correlation between PA and mental health. Second, all studies adopted subjective PA measures, and thus were subjected to social desirability bias or recall bias. In addition, this review included articles written in Chinese and English only; articles written in other languages may be omitted. Finally, this study focused on healthy residents, while vulnerable groups (e.g., pregnant women, people with disabilities, or patients) remain limited. Future studies are warranted to adopt intervention studies where PA baseline and changes are recorded and the mental health status is defined before and after the epidemic period, conduct objective PA measures, classify leisure and work-related PA, and focus on the vulnerable subpopulations to further explore the association of PA on mental health during the COVID-19 pandemic.

## 5. Conclusions

This review examines the relationship between PA and mental health during the pandemic. The study found that PA is related to mental health during the epidemic, and maintaining a certain amount of PA has a positive influence on mental health. However, physical inactivity or no exercise leads to a higher probability of mental health problems. Our findings have significant implications for addressing mental health issues during the current pandemic and future pandemics. Future studies adopting intervention studies where PA baseline and changes are recorded and the mental health status is defined before and after the epidemic period, conducting objective PA measures, classifying leisure and work-related PA, and focusing on the vulnerable subpopulations are warranted to further explore the association of PA on mental health during the COVID-19 pandemic.

## Figures and Tables

**Figure 1 ijerph-19-06584-f001:**
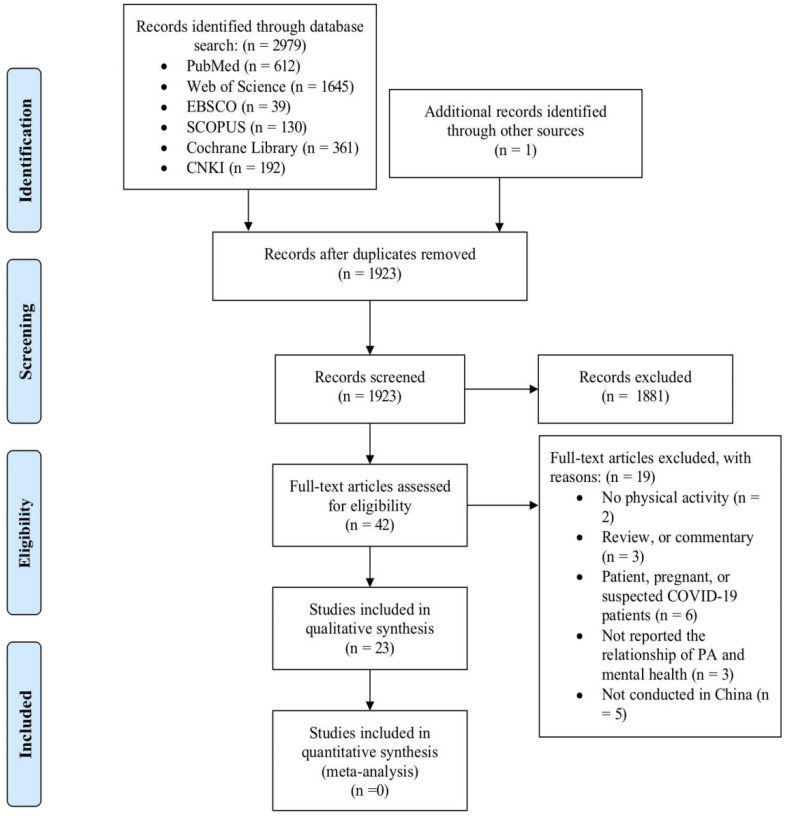
Study selection flow diagram.

**Table 1 ijerph-19-06584-t001:** Basic characteristics of the studies included in the review.

Study ID	First Author (Year)	City/Region/Country	Study Design	Sample Size	Age (Years)	Female (%)	Sample Characteristics	Statistical Model	Attrition Rate (%)	Setting
1	Chen Fangping, 2020 [15]	Guiyang	Cross-sectional	1036	6–15	48.7	Adolescents	Logistic regression	6.6	
2	Chen Gong, 2020 [16]	Nanjing	Cross-sectional	4750		65.2	College students	Multivariate logistic regression	3.2	
3	Deng, 2020 [6]	Wuhan	Cross-sectional	1607	18–22	35.2	College students	Linear regression	3.9	Rural, rural–urban, and urban
4	Fu, 2020 [17]	Wuhan	Cross-sectional	1242	18+	69.7	Residents	Multivariate logistic regression		Rural and urban
5	Hu, 2020 [18]	China	Cross-sectional	1033	18–60	48.2	Residents	Multivariate ordinal regression	32.4	Rural and urban
6	Li, 2020 [9]	China	Cross-sectional	3474			Adolescents	Pearson correlation analysis	8.3	Rural and urban
7	Lin Xiaogui, 2020 [19]	Fuzhou	Cross-sectional	1297		56.4	College students	Multinational logistic Regression	1.74	Rural and urban
8	Lin Jingyuan, 2020 [20]	China	Cross-sectional	625	20.17 ± 1.87	64.8	College students	Linear Regression	0.5	
9	Lu, 2020 [21]	Hechi	Cross-sectional	965	15.26 ± 0.46	42.4	Adolescents	Logistic regression	8.4	Rural and urban
10	Miao, 2020 [22]	China	Cross-sectional	3009	29.08 ± 7.45	50	WeChat users	Ologit model	62.5	
11	Qi, 2020 [23]	China	Cross-sectional	645	31.8 ± 8.6	61.2	Adults	ANOVA		Suburban and urban
12	Qin, 2020 [24]	China	Cross-sectional	12107	18–80	53.5	Adults	ANOVA		Rural and urban
13	Xiang, 2020 [25]	China	Cross-sectional	1396	20.68 ± 1.84	36.9	College students	Linear regression	1.8	
14	Zhang Xindan, 2020 [26]	China	Cross-sectional	706		41.5	College students	Multivariate logistic regression		
15	Zhang Xinxin, 2020 [27]	Yanan	Cross-sectional	9979	11.63 ± 1.23	48.58	Children and adolescents	General linear regression		
16	Zhang Yao, 2020 [13]	China	Longitudinal	66	20.70 ± 2.11	62.12	College students	Mixed-effect model and generalized additive model		Rural and urban
17	Zhou Jiaojiao, 2020 [28]	China	Cross-sectional	4805	11–18	100	Female adolescents	Multivariate logistic regression	4.9	
18	Zhou Jie, 2020 [29]	Nanjing, Suzhou	Cross-sectional	3248	19.64 ± 1.07	62.1	College students	ANOVA	9	
19	Chen, 2021 [14]	China	Longitudinal	9554 3886	11–20	52.1 56.7	Adolescents	Multivariable logistic regression		
20	Chi, 2021 [30]	Hechi	Cross-sectional	1794	15–18	43.8	Junior middle school students	Generalized linear models	3.3	Rural and urban
21	Kang, 2021 [31]	Yanan	Cross-sectional	4898	16.3 ± 1.3	52	Adolescents	Mixed liner regression	7.9	Rural and urban
22	Wu, 2021 [32]	Wuhu	Cross-sectional	2702	20.5 ± 0.9	74.9	College students	Multivariate logistic regression	3.2	Rural and urban
23	Xiao, 2021 [33]	Southwest China	Cross-sectional	1680		48.7	Adolescents of grades 7 to 12	Hierarchical regression analysis		Urban

**Table 2 ijerph-19-06584-t002:** Measures of physical activity and mental health in the studies included in the review.

Study ID	First Author	Type of Physical Activity Measure	Detailed Measure of Physical Activity	Type of Mental Health Measure		
				Anxiety	Depression	Other Mental Health Variables
1	Chen Fangping, 2020 [15]	Self-reported questionnaire	Regular physical exercise	Screen for Child Anxiety Related Disorders	Depression Self-Rating Scale for Children	
2	Chen Gong, 2020 [16]	Self-reported questionnaire	Physical exercise frequency	SAS		
3	Deng, 2020 [6]	Self-reported questionnaire WeChat’s pedometer	1. Exercise habits 2. Frequency and duration of exercise 3. Preferred sports 4. Pedometer steps	DASS-21	DASS-21	DASS-21
4	Fu, 2020 [17]	Self-reported questionnaire	Exercise habits	GAD-7	PHQ-9	Athens Insomnia Scale; Simplified Coping Style Questionnaire
5	Hu, 2020 [18]	IPAQ	1. Frequency of MVPA 2. Duration of PA			GWS
6	Li, 2020 [9]	IPAQ	1. PA level 2. Sedentary Time			POMS
7	Lin Xiaogui, 2020 [19]	Self-reported questionnaire	1. PA attitude 2. PA duration 3. PA frequency 4. PA intensity	GAD-7	PHQ-9	
8	Lin Jingyuan, 2020 [20]	IPAQ-Short Form	1. PA frequency and duration for vigorous-intensity activities, moderate-intensity activities, and walking 2. Sedentary Time 3. MET-minutes/week		CES-D	
9	Lu, 2020 [21]	IPAQ-Short Form	1. PA frequency and duration for vigorous-intensity activities, moderate-intensity activities, and walking 2. Sedentary Time	GAD-7	PHQ-9	YSIS; FCV-19S
10	Miao, 2020 [22]	Self-reported questionnaire	Physical exercise			Self-reported questionnaire
11	Qi, 2020 [23]	IPAQ-Short Form	1. PA frequency and duration for vigorous-intensity activities, moderate-intensity activities, and walking 2. Sedentary Time			1. The 10-item Perceived Stress Scale 2. SF-8
12	Qin, 2020 [24]	IPAQ-Short Form	1. PA frequency and duration for vigorous-intensity activities, moderate-intensity activities, and walking 2. Sedentary Time			The Positive and Negative Affect Schedule questionnaire of two 10-item scales
13	Xiang, 2020 [25]	IPAQ-Short Form	1. PA frequency and duration for vigorous-intensity activities, moderate-intensity activities, and walking 2. Specific types of PA: stretching, resistance training	SAS	SDS	
14	Zhang Xindan, 2020 [26]	Self-reported questionnaire	Physical exercise frequency			Self-reported questionnaire
15	Zhang Xinxin, 2020 [27]	IPAQ-Short Form	PA frequency and duration Sedentary Time			POMS
16	Zhang Yao, 2020 [13]	IPAQ-Short Form	PA frequency and duration Sedentary Time	DASS-21	DASS-21	1. Pittsburgh Sleep Quality Index 2. DASS-21 3. Buss–Perry Aggressive Questionnaire
17	Zhou Jiaojiao, 2020 [28]	Self-reported questionnaire	Physical exercise duration		CES-D	
18	Zhou Jie, 2020 [29]	Self-reported questionnaire	1. PA duration 2. PA frequency 3. PA intensity	SAS		
19	Chen, 2021 [14]	Self-reported questionnaire	Duration of exercise	GAD-7	CES-D	
20	Chi, 2021 [30]	IPAQ-Short Form	PA level	GAD-7	PHQ-9	FCV-19S; YSIS
21	Kang, 2021 [31]	IPAQ-Short Form	PA frequency and duration for vigorous-intensity activities, moderate-intensity activities, and walking Sedentary Time			POMS
22	Wu, 2021 [32]	Self-reported questionnaire	Physical exercise frequency	SAS	SDS	
23	Xiao, 2021 [33]	Leisure-Time Exercise Questions	1. PA level 2. PA frequency			POMS

Notes: PA, physical activity; IPAQ, the International Physical Activity Questionnaire; GAD-7, the 7-item Generalized Anxiety Disorder; SAS, Self-Rating Anxiety Scale; DASS-21, Depression Anxiety Stress Scale, with 21 self-reported items; PHQ-9, Patient Health Questionnaire 9; SDS, Self-Rating Depression Scale; POMS, the Profile of Mood States; CES-D, the Center for Epidemiological Studies Depression Scale; YSIS, Youth Self-Rating Insomnia Scales; FCV-19S, Fear of COVID-19 Scale; GWS, the General Wellbeing Schedule.

**Table 3 ijerph-19-06584-t003:** Estimated effects of physical activity on mental health in the studies included in the review.

Study ID (Year)	First Author	Estimated Effects of PA on Mental Health			Main Findings of Study
		Anxiety	Depression	Other Mental Health Variables	
1	Chen Fangping, 2020 [15]	Regular physical exercise was associated with anxiety significantly.	Regular physical exercise was associated with depression significantly.		Physical exercise was associated with both depression and anxiety.
2	Chen Gong, 2020 [16]	Physical exercise 1–2 times/week and anxiety: − (OR = 0.64, 95%CI = 0.50, 0.82; *p* < 0.01) Physical exercise 3–4 times/week and anxiety: − (OR = 0.60, 95%CI = 0.47, 0.78; *p* < 0.01) Physical exercise 5–6 times/week and anxiety: − (OR = 0.55, 95%CI = 0.41, 0.75; *p* < 0.01) Physical exercise more than 6 times/week and anxiety: − (OR = 0.49, 95%CI = 0.36, 0.68; *p* < 0.01)			High frequency of physical exercise can help to reduce students’ anxiety.
3	Deng, 2020 [6]	Exercise regularly and anxiety: − (B = −0.700, t = −5.636, *p* < 0.001) Maintained exercise habits and anxiety: − (B = −1.211, t = −6.988, *p* < 0.001) Exercised more than 1 to 2 times/week and anxiety: − (B = −0.089, t = −3.124, *p* = 0.002) With >2000 average pedometer steps and anxiety: −	Exercise regularly and depression: − (B = −1.257, t = −7.962, *p* < 0.001) Maintained exercise habits and depression: − (B = −2.017, t = −9.171, *p* < 0.001) Exercised more than 1 to 2 times/week and depression: − (B = −0.112, t = −3.946, *p* < 0.001) Exercised >1 h and depression: − (B = −0.588, t = −3.248, *p* = 0.001) With >2000 average pedometer steps and depression: −	Exercise regularly and stress: − (B = −1.013, t = −6.211, *p* < 0.001) Maintained exercise habits and stress: − (B = −2.198, t = −9.788, *p* < 0.001) Exercised more than 1 to 2 times/week and stress: − (B = −0.084, t = −2.949, *p* = 0.003) Exercised >1 h and stress: − (B = −0.503, t = −2.708, *p* = 0.007) With >2000 average pedometer steps and stress: −	Mental status was significantly correlated with regular exercise and sufficient exercise duration.
4	Fu, 2020 [17]	No exercise and anxiety: + (OR = 1.45, 95%CI = 1.08, 1.93; *p* = 0.013)	No exercise and depression: + (OR = 1.71, 95%CI = 1.28, 2.29; *p* = 0.000)	No exercise and sleep disorder: + (OR = 1.85, 95%CI: 1.38–2.47; *p* = 0.000) No exercise and passive coping style: + (OR = 1.71, 95%CI: 1.29–2.27; *p* = 0.000)	Not exercising was a common risk factor for anxiety, depression, sleep disorder, and passive coping style.
5	Hu, 2020 [18]			Inactive leisure-time PA and lower subjective well-being: + (OR = 1.16, 95%CI: 1.02–1.48;)	Both unhealthy lifestyle behaviors and negative lifestyle changes were associated with lower subjective well-being.
6	Li, 2020 [9]			*For adolescents in urban areas:* PA level and anger: − (r = −0.054, *p* < 0.05) PA level and emotional state: − (r = −0. 053, *p* < 0.05) Medium PA level and positive emotions: + Medium PA level and energetic emotions: + *For adolescents in rural areas:* PA level and emotions: 0	During the epidemic period at home, adolescents should do more moderate PA to maintain a healthy and positive mood.
7	Lin Xiaogui, 2020 [19]	*Mild anxiety:* Not willing to start exercising and mild anxiety: + (OR = 3.36, 95%CI = 1.42, 7.97; *p* < 0.05) Not yet, but willing to start exercise in the next 2 months and mild anxiety: + (OR = 4.07, 95%CI = 1.90, 8.74; *p* < 0.05) Occasionally exercise now, willing to start in the next 1 month and mild anxiety: + (OR = 2.65, 95%CI = 1.39, 5.05; *p* < 0.05) *Severe anxiety:* No exercise now and no intention to exercise regularly within the next 2 months and severe anxiety: + (OR = 15.61, 95%CI = 1.42, 170.98; *p* < 0.05).	*Mild* *depression:* Not willing to start exercising and mild depression: + (OR = 5.66, 95%CI = 2.38, 13.51; *p* < 0.05) Not yet, but willing to start exercise in the next 2 months and mild depression: + (OR = 5.97, 95%CI = 2.74, 13.00; *p* < 0.05) Occasionally exercise now, willing to start in the next 1 month and mild depression: + (OR = 3.43, 95%CI = 1.76, 6.68; *p* < 0.05) Exercise regularly but lasted less than 2 months and mild depression: + (OR = 2.22, 95%CI = 1.13, 4.37; *p* < 0.05) *Moderate depression:* No exercise and moderate depression: + (OR = 1.16, 95%CI = 0.34, 3.98; *p* < 0.05) No intention to exercise regularly and moderate depression: + (OR = 3.86, 95%CI = 1.09, 13.68; *p* < 0.05) *Severe depression:* No intention to exercise and severe depression: + (OR = 14.45, 95%CI = 1.05, 198.26; *p* < 0.05)		College students’ physical exercise will affect their mental health.
8	Lin Jingyuan, 2020 [20]		MET minutes/week in moderate intensity PA and depression: − (*p* < 0.05) Vigorous and walking scores and depression: 0 The depression–PA association was only moderated by the “low” activity level group in terms of categorical scores across gender groups.		Moderate-intensity PA is beneficial for reducing depression risk among college students at a low activity level.
9	Lu, 2020 [21]	High sitting time and anxiety: + (OR = 1.47, 95%CI = 1.11, 1.94; *p* < 0.01) High PA time and low sitting time and anxiety: − (OR = 0.65, 95%CI = 0.43, 0.97; *p* < 0.05)	High sitting time and depressive symptoms: + (OR = 1.65, 95%CI = 1.26, 2.16; *p* < 0.001) High PA time and depressive symptoms: − (OR = 0.71, 95%CI = 0.54, 0.93; *p* < 0.05) High PA time and low sitting time and depressive symptoms: − (OR = 0.43, 95%CI = 0.30, 0.63; *p* < 0.001)	High sitting time and insomnia symptoms: + (OR = 1.79, 95%CI = 1.36, 2.37; *p* < 0.001) High PA time and insomnia symptoms: − (OR = 0.71, 95%CI = 0.54, 0.95; *p* < 0.05) High PA time and low sitting time and insomnia symptoms: − (OR = 0.40, 95%CI = 0.27, 0.59; *p* < 0.001)	Moving more and sitting less were good for sleep and mental health in Chinese adolescents during the pandemic.
10	Miao, 2020 [22]			Physical exercise and self-rated health: + (β = 0.37, *p* < 0.001) Physical exercise and subjective well-being: + (β = 0.50, *p* < 0.001)	Active participation in physical exercise during the epidemic can improve physical and mental health and subjective well-being.
11	Qi, 2020 [23]			PA and perceived stress levels: − (r = −0.1, *p* = 0.002)	PA participation and perceived stress are significantly related among Chinese adults during the COVID-19 pandemic.
12	Qin, 2020 [24]			Vigorous PA and positive affect scores: + (*p* < 0.0001)	Individuals engaging in vigorous PA had better emotional states while the group engaging in only light activity showed the opposite trend.
13	Xiang, 2020 [25]	High level PA and anxiety: − (β = −0.121, *p* < 0.001) Stretching training and anxiety: − (β = −0.082, *p* < 0.01) Resistance training and anxiety: − (β =−0.058, *p* < 0.05)	Moderate-level PA and depression: − (β = −0.095, *p* < 0.01) High-level PA and depression: − (β = −0.179, *p* < 0.001) Stretching training and depression: − (β = −0.122, *p* < 0.001) Resistance training and depression: − (β = −0.131, *p* < 0.001)		Moderate and high levels of PA, as well as specific types of PA, such as stretching and resistance training, were protective factors against anxiety or depression among the college students.
14	Zhang Xindan, 2020 [26]		Indoor exercise frequency and depression: − (*p* < 0.05)		Maintaining exercise regularity has a positive effect on the physical and mental health of the students who are quarantined at home during the COVID-19 pandemic.
15	Zhang Xinxin, 2020 [27]		Moderate-level PA and depression: − (*p* < 0.01) High-level PA and depression: − (*p* < 0.01)	Moderate-level PA and (confusion, anger, and fatigue): − (*p* < 0.01) High-level PA and (confusion, anger, and fatigue): − (*p* < 0.01) Moderate-level PA and vigor: + (*p* < 0.01) High-level PA and vigor: + (*p* < 0.01)	PA was related to the mood states of children and adolescents, and lower PA levels showed higher scores in negative mood states.
16	Zhang Yao, 2020 [13]		PA and depression: − (*p* < 0.05)	PA and negative emotions: − (*p* < 0.05) PA and global DASS score: − (β = −0.12, 95%CI= −0.22, −0.010; *p* < 0.05) PA and stress: 0 (*p* > 0.05)	Taking suitable amounts of daily PA is a possible mitigation strategy for improving mental health. The dose–response curve between negative emotions and PA exhibited a U-shaped relationship.
17	Zhou Jiaojiao, 2020 [28]		Physical exercise duration/day of < 30 min and depression: + (OR = 1.641, 95%CI = 1.455, 1.850; *p* < 0.001)		Less physical exercise was significantly associated with higher risk of depression.
18	Zhou Jie, 2020 [29]	Regularly exercised at home and anxiety: − (*p* < 0.01)			The anxiety degree of students who regularly participate in physical exercise is lower than that of students who do not regularly exercise.
19	Chen, 2021 [14]	Exercise duration of 30–60 min/day and anxiety: − (OR = 0.73, 95%CI = 0.67, 0.80) Exercise duration ≥60 min/day and anxiety: − (OR = 0.84, 95%CI = 0.73, 0.98)	Exercise duration of 30–60 min/day and depression: − (OR = 0.64, 95%CI = 0.60, 0.70) Exercise duration ≥60 min/day and depression: − (OR = 0.67, 95%CI = 0.59, 0.77)		An exercise duration ≥30 min/day was negatively associated with depression and anxiety.
20	Chi, 2021 [30]	Moderately active physically and anxiety symptoms: − (β = −0.16 [−0.27, −0.05], *p* = 0.005) Highly active physically and anxiety symptoms: − (β = −0.15 [−0.25, −0.05], *p* = 0.004)	Moderately active physically and depressive symptoms: − (β = −0.16 [−0.26, −0.06], *p* = 0.002) Highly active physically and depressive symptoms: − (β = −0.17 [−0.27, −0.08], *p* < 0.001)	Being highly active physically was associated with lower level of insomnia symptoms (β = −0.05 [−0.10, −0.01], *p* = 0.020).	Both moderately and highly active (PA) levels are associated with lower level of depressive and anxiety symptoms, while highly active PA level was significantly associated with lower level of insomnia symptoms.
21	Kang, 2021 [31]			Higher PA and negative mood states: − (*p* < 0.05) Sedentary time and mood state: 0	Higher levels of PA were associated with better mood states.
22	Wu, 2021 [32]	Physical exercise 1–2 times/week and anxiety: − (OR = 0.67, 95%CI = 0.48, 0.94; *p* = 0.02).	Physical exercise 1–2 times/week and depression: − (OR = 0.73, 95%CI = 0.58, 0.91; *p* < 0.01) Physical exercise 3–4 times/week and depression: − (OR = 0.65, 95%CI = 0.50, 0.86; *p* < 0.01)		Moderate physical exercise habit is a protective factor of college students’ depression, but excessive exercise will affect the level of depression and anxiety of college students.
23	Xiao, 2021 [33]			PA ≥ 150 min/week and negative mood: − (β = −10.98, *p* < 0.01) Adding PA participation and mood disturbance score: − (β = –0.03, *p* < 0.001)	Promoting PA and decreasing screen time among adolescents during school closure is an effective way to minimize negative mood.

Notes: PA, physical activity; Correlation: + positively, − negatively, 0 insignificantly.

## Data Availability

Not applicable.

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
