# Peer review of "The Impact of Physical Activity on Mental Health during COVID-19 Pandemic in China: A Systematic Review"

_ijerph, 2022, doi:10.3390/ijerph19116584_

Round 1
Reviewer 1 Report
This systemic review highlights the need to invest further in studying the links between COVID-19 pandemic and mental health. Physical activity is one of the factors that might have a positive impact on mental health. However, as mentioned by the authors, the findings of the review do not reveal causality: physical health as a protector. I suggest to emphasise the need to focus on intervention studies where people's physical activity baseline is recorded and is maintained/improved through the period. The Mental Health status should be defined before and after.
Also, important to mention the ways to overcome classification bias due to leisure and work-related physical activity.
I would ask the authors to highlight these in the Conclusion section.
Also, please proofread again (e.g. line 70 "health" to be changed to "healthy", line 80 item #4 appears twice - change the last one to 5).
Reviewer 2 Report
This paper systematically reviewed and confirmed the positive relationship between physical activity and mental health during the pandemic in China. This is a generally carefully constructed paper and I recommend for publication after a minor revision.
In lines 11-12, the “health behavior intervention” is duplicated and awkward in writing. The two sentences should be re-written.
In line 20, the “..U-shaped relationship, and rural-urban difference between PA and emotional state” is not clear in meaning. It should be re-written.
In the tables, the first author column is redundant and not informative. For improved readability, this column may be replaced by the citation number of the associated paper.
In the footnote of Table 2, acronyms should be explained by words in lower case unless in special cases, e.g., for DASS-21, it should be “depression anxiety stress scale”.
Reviewer 3 Report
Introduction
It could be more concise, some repetitions can be avoided, see lines 52-53 and 57-58.
Materials and methods
Study design was well established and described.
It would be useful to introduce reference about the "National Institutes of Health’s Quality Assessment Tool for Observational Cohort and Cross-Sectional Studies", used by authors to assess quality of included studies
Results
Inclusion criteria should take into account that rate of physical activity and risk of mental problems is different in young people (children/adolescents, who represent the majority of patients included) than in older people (until 80 years of age): this can represent a bias in the interpretation of results. Consider to restrict analysis to young population. Furthermore, for review purposes, studies performed over 2020 and 2021 were retreived. Measures to contain pondemic in China were very severe in 2020 with a strict lock-down, but much less restrictive in 2021. This difference, alone, may have had an influence on residents mental health.
Tab 3 appears too long and difficult to read, information are not schematic and could be better reported in a paragraph.
Tab 4 could be included as supplementary material
Discussion
Authors claim a greater effects of lower physical activity during the COVID19 pandemic, on mental health in women compared to men, however, this conclusion is not justified in the results section. Also the sentence reported in lines 325-27 is not justified by results. Please provide further details.
Reviewer 4 Report
Check that the abbreviations used are defined before, for example in the abstract they use PA and have not defined it.
